# Acceptability and feasibility of tests for infection, serological testing, and photography to define need for interventions against trachoma

Kristen K. Renneker[1,2]*, Tara B. Mtuy[1,3], George Kabona[4], Stephen Gabriel Mbwambo[4], Patrick Mosha[2], Jeremiah Mepukori Mollel[2], PJ Hooper[2], Paul M. Emerson[2], T. Deirdre Hollingsworth[5], Robert Butcher[1], Anthony W. Solomon[6], Emma M. Harding-Esch[1]

1 Clinical Research Department, Faculty of Infectious and Tropical Diseases, London School of Hygiene & Tropical Medicine, London, United Kingdom, 2 International Trachoma Initiative, The Task Force for Global Health, Decatur, Georgia, United States of America, 3 Kilimanjaro Christian Medical Centre, Moshi, Tanzania, 4 National Neglected Tropical Diseases Control Programme, Preventive Services Department, Ministry of Health, Dodoma, Tanzania, 5 Big Data Institute, Li Ka Shing Centre for Health Information and Discovery, University of Oxford, Oxford, United Kingdom, 6 Global Neglected Tropical Diseases Programme, World Health Organization, Geneva, Switzerland

* krenneker@taskforce.org

**Data Availability Statement:** Data access can be obtained by emailing the International Trachoma

## Abstract

### Background

Trachoma causes blindness due to repeated conjunctival infection by *Chlamydia trachomatis* (*Ct*). Transmission intensity is estimated, for programmatic decision-making, by prevalence of the clinical sign trachomatous inflammation—follicular (TF) in children aged 1–9 years. Research into complementary indicators to field-graded TF includes work on conjunctival photography, tests for ocular *Ct* infection, and serology. The perceived acceptability and feasibility of these indicators among a variety of stakeholders is unknown.

### Methodology

Focus group discussions (FGDs) with community members and in-depth interviews (IDIs) with public health practitioners in Tanzania were conducted. FGDs explored themes including participants' experience with, and thoughts about, different diagnostic approaches. The framework method for content analysis was used. IDIs yielded lists of perceived strengths of, and barriers to, implementation for programmatic use of each indicator. These were used to form an online quantitative survey on complementary indicators distributed to global stakeholders via meetings, mailing lists, and social media posts.

### Results

Sixteen FGDs and 11 IDIs were conducted in October–November 2022. In general, all proposed sample methods were deemed acceptable by community members. Common themes included not wanting undue discomfort and a preference for tests perceived as

Initiative (ITI)'s Data and Analytics Team at itidat@taskforce.org.

**Funding:** Study activities were funded by the International Trachoma Initiative. KKR, PJH, and PME are employees of ITI, a program of The Task Force for Global Health, which receives an operating budget and research funds from Pfizer Inc., the manufacturers of Zithromax (azithromycin). EMHE receives salary support from ITI, and PM and JMM were contractors of ITI for the duration of field activities in Tanzania. AWS is a staff member of the World Health Organization. RBs salary was funded by the Wellcome Trust (206275/Z/17/Z). The funders had no role in study design, data collection and analysis, decision to publish, or preparation of the manuscript.

**Competing interests:** All authors have declared that no competing interests exist.

accurate. Health workers noted the importance of community education for some sample types. The online survey was conducted in April–May 2023 with 98 starting the questionnaire and 81 completing it. Regarding barriers to implementing diagnostics, the highest agreement items related to feasibility, rather than acceptability. No evidence of significant differences was found in responses pertaining to community acceptability based on participant characteristics.

## Conclusions

All of the indicators included were generally deemed acceptable by all stakeholders in Tanzania, although community education around the benefits and risks of different sample types, as well as addressing issues around feasibility, will be key to successful, sustainable integration of these indicators into trachoma programs.

### Author summary

Trachoma is a disease that causes blindness through conjunctival infection with the bacterium *Chlamydia trachomatis*. Trachoma is targeted for global elimination by 2030. To know whether population-level interventions are required, we must know how intensely conjunctival *C. trachomatis* is being transmitted in a population. The current proxy recommended by the World Health Organization is prevalence of a clinical sign of active (inflammatory) trachoma: trachomatous inflammation—follicular. However, this indicator has several drawbacks. Policy-makers are considering the utility of a number of complementary indicators, including conjunctival photography and tests for infection and serology.

We sought the opinions of different stakeholders to determine the acceptability and feasibility of complementary indicators for use in trachoma programs. In Tanzania, we undertook focus group discussions with community members and in-depth interviews with public health practitioners. We also conducted an online survey of global stakeholders. We found that all the proposed test types were acceptable to stakeholders in Tanzania; common themes included not wanting undue discomfort and a preference for test types perceived to be accurate. Community education and building trust were deemed critical. From the online survey, the most agreed-upon barriers to implementation of each method were related to concerns about feasibility, rather than acceptability.

## Introduction

Trachoma is a neglected tropical disease (NTD) that causes blindness as a result of repeated conjunctival infection by the bacterium *Chlamydia trachomatis* (*Ct*). [1] Transmission intensity is estimated, for programmatic decision-making, via field grading of the clinical sign trachomatous inflammation—follicular (TF). A major goal of the World Health Organization (WHO) Alliance for the Global Elimination of Trachoma is the reduction of TF prevalence in 1–9-year-olds ($TF_{1-9}$) to <5% in all formerly endemic districts worldwide, contributing to the achievement of global elimination of trachoma as a public health problem. [2] A key component of the WHO-recommended SAFE (surgery, antibiotics, promotion of facial cleanliness and environmental improvement) strategy that is designed to be used to achieve this target is

mass drug administration (MDA) of antibiotics to clear infection. The number of MDA rounds is dependent on the category of $TF_{1-9}$ within an evaluation unit (generally equivalent to a district) as measured by population-based prevalence surveys [3] on the assumption that $TF_{1-9}$ is well-correlated with prevalence of $Ct$ infection.

However, $TF_{1-9}$ has several limitations as the sole indicator for trachoma. First, the relationship between $TF_{1-9}$ and infection is not predictable, [4] especially post-MDA, [5] and the reduction in $TF_{1-9}$ often lags behind the reduction in $Ct$ infection. [6] Second, conjunctival follicles (the features that lead to a diagnosis of TF) are not necessarily specific to trachoma, sometimes occurring due to other causes. [7] Furthermore, in areas with low prevalence of trachoma, it becomes both costlier to train field graders [8] and more difficult for graders to pass the inter-grader agreement test. [9–11]

There has been ongoing research into indicators that could be alternative or complementary to TF, including conjunctival photography, [12] tests for ocular $Ct$ infection, [5] and serology to assess an individuals' previous exposure to $Ct$ infection by detecting antibodies to the pathogen. [13] It is important to note that each of these indicators measures a different signal: clinical grading (both field-based and through photography) measures signs of inflammation, infection tests are a direct measure of current infection with $Ct$, and serology measures a history of previous exposure to the pathogen. Therefore, each of these indicators could provide different measures of the public-health threat caused by trachoma; i.e., indicators should be considered "complementary" rather than "alternative" to each other.

Following a WHO meeting in 2016 reviewing existing data, it was concluded that there was insufficient evidence to support routine use of $Ct$ infection or serology tests to inform trachoma elimination programs. [14] However, a WHO informal workshop in December 2021 supported the use of age-stratified data on $Ct$ infection and serological data on the presence of anti-$Ct$ antibodies for decision-making in districts where $TF_{1-9}$ either remains at or above the 5% elimination threshold at the second impact survey (districts with "persistent" TF) following MDA interventions, or falls <5% but subsequently returns to ≥5% during the surveillance period after cessation of MDA (districts with "recrudescent" TF). [15]

WHO is now producing guidelines on the use of serology in trachoma elimination programs. As part of the guideline development process, the Guideline Development Group must consider both the acceptability of the proposed intervention and the values and preferences of the people affected by the recommendations, which will partially determine the strength of any recommendations. [16] There have been many studies conducted that incorporate conjunctival photography, infection testing, and serology and anecdotally, refusals of these test types seem to be rare; however, response rates are not often reported. Additionally, studies are often done in places with established programs which may bias toward favorable response rates as these communities have been well-sensitized to trachoma research activities.

Research to date has indicated the value complementary indicators could have for trachoma surveillance purposes. A literature review assessing post-elimination surveillance systems for multiple diseases, including trachoma, suggested that opportunities for inclusion of infection testing and serological markers in trachoma elimination settings continue to be explored. [17] A 2019 survey of stakeholders found that the top-cited barrier to trachoma eradication is inadequate surveillance tools and systems to monitor for recrudescence. [18] While eradication is a different goal than elimination as a public health problem, the need for practical strategies to measure trachoma after the cessation of MDA remains.

With 2030 as the target for global elimination of trachoma as a public health problem, [19] a better understanding of the acceptability and feasibility of complementary indicators for trachoma detection by all stakeholders is urgently needed. In this study, we aimed to determine the opinion of a wide variety of stakeholders through a mixed-methods approach: a qualitative

study of community members and public health practitioners in Tanzania, followed by a quantitative online global survey of stakeholders.

## Methods

### Qualitative study in Tanzania

**Ethics statement.**   The qualitative study was approved by the London School of Hygiene & Tropical Medicine Research Ethics Committee (Ref: 28028), the Task Force for Global Health ethical review body, and Tanzania's National Institute of Medical Research (Ref: NIMR/HQ/R.8a/Vol.IX/4123). Work was conducted in collaboration with Tanzania's health ministry through the National NTD Control Programme, region- and district-level health departments, and village leaders. Written, informed consent was obtained from all participants prior to their inclusion by trained research assistants fluent in the local languages (Swahili and/or Maa) with an impartial witness present for illiterate participants. Permission to record the focus group discussions (FGDs) and in-depth interviews (IDIs) and publish quotes was obtained at enrollment.

**Setting/sampling.**   Tanzania was chosen based on the support of its health ministry, history and strength of its trachoma program, presence of districts with persistent and recrudescent TF as well as districts having met the elimination threshold, and access to the "Maasai corridor", an area spanning the Kenya-Tanzania border with high levels of persistent and recrudescent TF among pastoralists including the Maasai ethnic group. FGDs with community members were conducted in four districts in Tanzania, consisting of two districts with $TF_{1-9}$ <5% and two with $TF_{1-9}$ ≥5% at most recent survey, chosen to be representative of pre- and post-elimination settings. Districts were chosen in three selected regions (Dodoma, Singida, and Arusha) to ensure a diversity of trachoma program experience, geographic location, and ethnic group. The current $TF_{1-9}$ category in Tanzania and the four selected districts are shown in Fig 1. Within each chosen district, two villages were randomly selected for the FGDs by assembling a list of all villages in each district, assigning each village a random number, and selecting the two villages with the lowest randomly generated numbers.

For the FGDs, community members aged 18–60 years who had experience caring for children were eligible to participate. Participants were identified using a snowball sampling approach, [21] with index participants chosen by the village leader. Snowball sampling continued until the target group size (n = 10) for each FGD was reached. Separate FGDs were held with men and women.

IDIs were conducted with public health practitioners in Tanzania. We sought interviews with a wide variety of public health practitioners, including laboratory personnel, NTD program officers, and non-governmental organization (NGO) program staff at all levels. Participants were selected based on experience working in NTD programs and their availability, with a target of 2–3 respondents for each of the three regions selected for FGDs, plus practitioners at the national level who were based in Dodoma.

**Data collection.**   FGDs and IDIs were conducted by trained research assistants in the participants' preferred language using pre-designed topic guides (S1 File), exploring themes such as previous experience with each of the test types: TF status via field grading and photography, infection testing, and serology, as well as (for community members) thoughts about the different sample collection methods: field grading, conjunctival photography, eye swabs, and dried blood spots, and (for public health practitioners) thoughts about the feasibility of each test type. In the FGDs, visual aids were used to help participants understand the different sample collection methods. Sessions were recorded using an electronic audio recorder, transcribed by

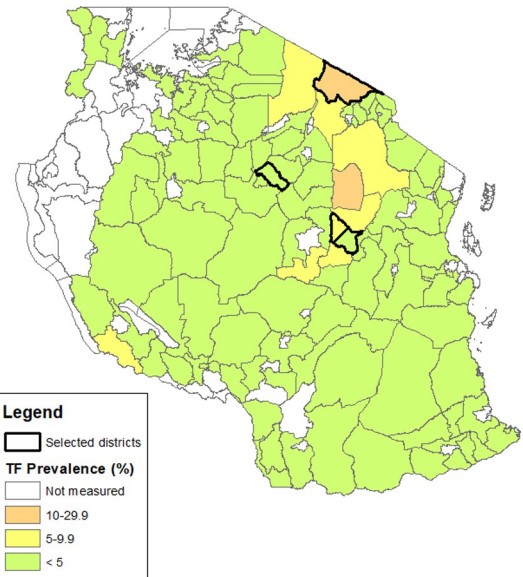

**Fig 1. Map of the four districts selected for the study and their current trachomatous inflammation—follicular (TF) prevalence category in children 1–9-years-old, Tanzania.** The boundaries and names shown and the designations used on this map do not imply the expression of any opinion whatsoever on the part of the authors, or the institutions with which they are affiliated, concerning the legal status of any country, territory, city or area or of its authorities, or concerning the delimitation of its frontiers or boundaries. Prevalence category source: Trachoma Atlas. [20] Basemap provided by GADM.

a native speaker and then translated by them into English. Translations were checked for accuracy by the researcher who carried out data collection.

**Data analysis.** Transcripts were entered into the qualitative data management software Nvivo [22] and coded based on both an *a priori* list of codes pertaining to each test type (S2 File) as well as broader emergent themes. The framework method for content analysis [23] was used to chart data into a framework matrix. For the FGDs, comparative analysis was done to compare responses between communities based on gender and setting ($TF_{1-9} \geq 5\%$ or $TF_{1-9} < 5\%$). The coding framework and analysis plan were reviewed by a second researcher prior to finalization of analysis. For the IDIs, responses pertaining to strengths and barriers for each of the sample types were used to formulate the stakeholder survey questionnaire.

## Quantitative survey of stakeholders

**Ethics statement.** The quantitative survey was approved by the London School of Hygiene & Tropical Medicine Research Ethics Committee (Ref: 28589) and the Task Force for Global Health ethical review body. Informed consent was obtained electronically in the preferred language of the participant, from the four languages available: English, Spanish, French, and Portuguese. All questions (other than consent) were optional, and the survey responses were anonymous. At the end of the survey window, survey data were downloaded from SurveyMonkey [24] prior to deletion of the survey and data from the platform.

**Setting/sampling.** Stakeholders working in trachoma, including national program coordinators or other national program staff, members of NGOs, donor organizations, and academic organizations were asked to participate in an online survey to assess their agreement with the strengths and barriers to each test type identified by the IDIs with Tanzanian public health practitioners. Stakeholders were required to be over the age of 18 years. A list of email addresses of stakeholders meeting these inclusion criteria was assembled using publicly

available sources, such as organizational websites, publications, and meeting reports with participant lists. Snowball sampling was employed by asking key members of partner organizations to recommend additional stakeholders. We solicited responses from as many people as possible who met our criteria.

**Data collection.**   The online survey began during the annual meeting of the WHO Alliance for the Global Elimination of Trachoma by 2020 (GET2020) held in Istanbul, Turkïye, in April 2023. The GET2020 meeting convenes a variety of stakeholders; representatives from all trachoma-endemic, formerly-endemic, and suspected-endemic countries are invited, as well as members of related NGOs and academic or research organizations. The survey was hosted on SurveyMonkey. [24] A link to the survey was available via email sent to the previously assembled list of email addresses, through advertisements at the GET2020 meeting, and through a post on the social media platform X (formerly known as Twitter) from EMHE. Prior to the start of the survey window, the survey was piloted with colleagues with expertise in trachoma and online survey forms to gain feedback on the questionnaire and the ease of using the survey platform. The questionnaire (S3 File) included questions on the participant's age group, gender identity, organizational role, primary country/countries of trachoma work, level of agreement with the importance of various perceived barriers to implementation at the program level and strengths for different complementary indicators on a 5-point Likert scale (with the options of: Strongly Agree, Agree, Neither Agree nor Disagree, Disagree, Strongly Disagree), and identification of further knowledge gaps remaining, entered as free text, that may prevent implementation of these indicators for trachoma surveillance. The strengths and barriers presented in the questionnaire were obtained from the responses of IDI participants.

**Data analysis.**   Descriptive analysis was performed using R. [25] Since the total target population is unknown (due to the survey being publicly available via an online link) no response rate could be calculated. Frequency tables were created to determine the percent of respondents that strongly agreed or agreed with, neither agreed nor disagreed with, and disagreed or strongly disagreed with each proposed strength and barrier. The "strongly agree" and "agree" responses were combined and the "strongly disagree" and "disagree" responses were combined to determine the percent of respondents who agreed or disagreed with a statement, respectively. In order to determine if responses pertaining to community acceptability differed between national program staff and other stakeholders, responses pertaining to these barriers were analyzed on the basis of the participant's organizational and personal role (country staff, as defined as a respondent whose organizational role was "Government" and personal role was "Programmatic/Implementation", compared to all other stakeholders) by calculating the median and interquartile range (IQR) of responses to these questions. The 2-tailed Wilcoxon rank-sum test was used to calculate p-values comparing the responses of the two groups. Key knowledge gaps entered by respondents as free text responses were cleaned and aggregated.

## Results

### Qualitative study in Tanzania

Data collection took place in Tanzania from October to November 2022. A total of 160 community members participated in 16 FGDs, with ten people per FGD. The reported population of selected villages ranged from roughly 1,100 to 5,400. Equal numbers of men and women participated. Fourteen FGDs were conducted in Swahili and two were conducted in Maa. A total of 11 public health practitioners participated in IDIs. The roles of participants included current and former country-level staff, NGO staff, lab technicians, regional NTD coordinators, and one healthcare worker. The framework matrix used to assess community responses for each test type by gender and setting is provided as S4 File.

**TF field grading.** According to community members in Tanzania, key advantages of field grading of TF were the familiarity of this method (particularly noted in FGDs with women), the lack of pain or adverse consequences from this method, and the ability to receive results immediately. A perceived disadvantage of field grading of TF, mentioned in half of the FGDs, was the possibility of short-term discomfort or adverse consequences that may be caused by the strong light of torches used as part of the field grading process:

"*I think that [the] examination method of being graded with torches causes effects. For example, eyes turn red and keeps on shedding a lot of tears after a certain period of time. I recommend that they should innovate a new method for trachoma examination.*" (31–35-year-old woman, $TF_{1-9}$ <5% district)

In the IDIs, most of the public health practitioners who provided advantages of field grading of TF mentioned that this method is both familiar and acceptable to communities, which agrees with community feedback from the FGDs. Very few IDIs mentioned any disadvantages with field grading of TF, and most disadvantages related to general concerns with needing to train staff to conduct the examination, which is not necessarily specific to this method. None of the IDIs mentioned the possibility that communities may dislike this method due to fear of adverse consequences from the torch used as part of the examination.

**Conjunctival photography.** A majority of FGDs mentioned that a key advantage of conjunctival photography is a perceived increase in the accuracy of grading via photo versus field grading due to the ability for graders to have ample time:

"*It is possible that [TF field grading] won't be conducted as it is required because you are in [a] hurry but when you are taking a photo it is a good thing because a person who will examine that photo will have plenty of time to do that.*" (26–30-year-old man, $TF_{1-9}$ <5% district)

Other advantages of conjunctival photography mentioned by FGD participants were the familiarity with the concept of imaging for diagnostics, and the lack of pain and adverse consequences from this method. Each of these advantages was particularly often noted in the FGDs with men. A majority of FGDs (particularly with men) mentioned that a key disadvantage is a fear of long-term adverse consequences caused by the photo flash:

"*Some people are worried that when you are photographed you will go blind or your eye will be destroyed.*" (46–50-year-old man, $TF_{1-9}$ <5% district)

Several FGDs (particularly those with men) also indicated a belief that conjunctival photography is not as accurate as other methods:

"*A photo is not enough; we would like to get another examination method because the camera won't show the disease that I have in my eyes or in my eyelid clearly.*" (31–35-year-old woman, $TF_{1-9}$ <5% district)

None of the FGD participants mentioned that they had concerns related to confidentiality, usage of the photos, or the potential identifiability of photos taken.

In the IDIs, none of the respondents mentioned a high level of community acceptability of conjunctival photography as a specific strength of this method. Several IDI participants mentioned that community fears related to flash photography causing adverse consequences and mistrust that the entire face would be photographed (leading to concerns by community

members around the confidentiality of their and their children's images) may be potential barriers to this method. Several mentioned that education would be needed in order for communities to participate:

> "*The most important thing is to give education and sensitization. If you will just tell a parent that you want to take a photo, he/she might think you have [a] hidden agenda with the child.*" (Program coordinator)

In terms of feasibility, while there was an acknowledgement from a few IDI participants that the fragility of camera equipment may be a concern in the field, most indicated that this would not be a true barrier to implementation, especially given the benefits of conjunctival photography.

**Eye swabs.** A majority of FGDs mentioned increased perceived test accuracy (especially compared to field or photo grading) as a key advantage of tests for ocular *Ct* infection. Half of FGDs (particularly those with men) indicated that an advantage of the test for ocular *Ct* infection is that this method is perceived as modern, high-tech, "professional", and/or legitimate (as it is approved by an authorizing body prior to use).

Most FGDs (particularly those with men) mentioned that a key disadvantage of eye swabs is a fear of long-term adverse consequences from the test procedure:

> "*I won't agree with the use of eye swab method, I will even tell my family members not to participate in that method because it has effects, it may leave you with a wound in the eye.*" (31–35-year-old-man, $TF_{1-9} \geq 5\%$ district)

In addition, half of FGDs (particularly those with men) mentioned a fear of pain from the test procedure, especially with regard to children. Several FGDs indicated that the complicated nature of the test (with the need for sample collection, storage, and testing) may cause test errors or failures.

From the IDIs, many respondents said that eye swabs would be acceptable to communities only with proper education as to the risks and benefits of participation. Some IDI respondents said that training would be needed in order for the sample collector to not harm the eyes of participants, with the importance of this being two-fold: to minimize harm to participants and increase participation rates:

> "*. . .if the trained personnel will not be careful when taking [the] sample, she/he can harm a child and these children can inform each other that it hurts when they are taking [the] sample and this will make all children not participate in [the] examination.*" (Lab technician)

In terms of feasibility, several IDI respondents said that a desire for samples to be processed at the local level (in order to increase turn-around time) is likely unfeasible due to a lack of current lab capacity at the district level. This was mentioned by both public health practitioners at the national and lower levels of administration.

**Blood spots.** Key advantages to blood spots mentioned in a majority of FGDs were the familiarity with blood testing and (particularly among FGDs with women) the lack of pain or adverse consequences from the testing procedure. Half of FGDs (particularly those with men) indicated that the high perceived accuracy of the results is an advantage to this method:

> "*There are many diseases that cannot be seen easily but when you take a blood sample to the lab it will be a perfect solution.*" (51–55-year-old man, $TF_{1-9} <5\%$ district)

Several FGDs mentioned that a disadvantage to blood spots is a lack of trust by participants that the sample will be used only to test trachoma; with fears especially around testing for HIV or other sexually transmitted infections without participants' consent. Many FGDs with participants who mentioned this fear indicated that it could be mitigated with community education, as community members had simply not been made aware that the presence of antibodies to trachoma could be tested for via blood spots prior to participating in the FGDs.

One FGD among the Maasai ethnic group mentioned a general dislike of blood testing in particular:

"*We, as Maasai people don't like blood testing most of the time. For example, HIV and TB are prevalent in Maasai society and when people are asked to test using blood samples, they run away.*" (46–50-year-old woman, $TF_{1-9} \geq 5\%$ district)

When IDI participants mentioned advantages of blood spots, community acceptability was not often specifically mentioned as a key strength of this method. When community acceptability was mentioned, the need for education was underscored. One IDI participant mentioned the potential for distrust related to sample use (specifically related to fears around HIV testing) among community members. Some IDI participants also mentioned the potential for community acceptability issues if community participants expect on-the-spot results (as they would receive for other types of blood tests they undergo as part of routine health care, such as malaria testing), and again underscored the need for community education as to the purpose of the test in the context of trachoma programming.

In terms of feasibility, key strengths mentioned by IDI participants included the familiarity of blood spot collection and testing by health systems and the ability for high throughput of these sample types, although several participants mentioned a lack of lab capacity to store and test samples at a local level as a potential issue. While familiarity of blood testing by health systems was often mentioned, IDI respondents did not mention the potential for integration of trachoma monitoring into existing health systems via the use of blood spots as a specific advantage of this method.

When asked, the majority of FGDs had at least one participant who would hypothetically participate in each sample collection method. No qualitative difference in responses between districts with a $TF_{1-9} \geq 5\%$ vs. $<5\%$ was detected (15 FGDs in districts with a $TF_{1-9} \geq 5\%$ vs. 15 FGDs in districts with a $TF_{1-9} <5\%$ with at least one respondent who said they would hypothetically participate in each method).

FGDs with men generally indicated more willingness to hypothetically participate in all of the testing methods compared to FGDs with women (16 FGDs with men vs. 14 FGDs with women with at least one respondent who said they would hypothetically participate in each method). While responses from FGDs with men and women were largely quite similar, FGDs with men often mentioned the importance of perceived accuracy, a lack of adverse consequences from the testing procedure, and the benefit of tests perceived as modern or high tech as important to determining the acceptability of the different methods. FGDs with women mentioned the importance of test familiarity and a lack of undue discomfort or harm from participating in the test procedure as important to determining the acceptability of the different methods.

## Quantitative survey of stakeholders

The online survey was live from April 24 to May 31, 2023. Responses from 98 people who consented to participate were received. Respondents reported living and working in 42 different

countries and represented each trachoma-endemic WHO Region (African Region, Eastern Mediterranean Region, Region of the Americas, South-East Asia Region, and the Western Pacific Region). A majority (60%) of respondents were men. Most respondents were between the ages of 35–49 years, worked for an NGO, and worked in a programmatic/implementation role (Table 1).

Eighty-one respondents answered questions beyond providing demographic information. Table 2 includes results for questions related to respondents' agreement with strengths and barriers to each indicator. Almost all (79, 98%) of respondents agreed that TF grading being WHO-recommended for diagnosing trachoma is a strength of field grading of TF. Besides this, the strengths of field grading of TF respondents most agreed with were that TF grading is familiar (71, 88%), acceptable to communities (69, 85%), harmless/does not cause side effects (66, 81%), and that participants are provided results on-the-spot (66, 81%). The most agreed-upon barrier indicated for field grading of TF was that it is a subjective measure that is prone to human error (65, 80%).

The most agreed-upon strength for conjunctival photography was that multiple graders can grade the same photo, increasing accuracy and helping to determine the status of borderline cases (61, 84%). In contrast, a third of respondents (24, 33%) said that a strength of conjunctival photography is that the ability to grade the photo away from the field would reduce field time. The most agreed-upon barrier to implementation of conjunctival photography is that it would increase the costs compared to TF field grading due to the need for camera equipment (43, 60%), followed closely by increased costs due to the need for increased personnel training and time (41, 57%).

For infection testing, the most agreed-upon strength was this indicator's ability to detect infection in the absence of clinical signs (55, 81%). The most agreed-upon barriers for this indicator all pertain to challenges around the feasibility of implementation, including the extra work required to plan for logistics of field collection and sample processing (68, 96%), the challenge posed by a desire for samples to be tested by the nearest lab (63, 89%), increased costs compared to field grading of TF posed by the need to store samples (62, 87%) and purchase

**Table 1. Self-reported demographic information of respondents to stakeholder survey.**

| Characteristic | n | % |
|---|---:|---:|
| **Gender** | | |
| Man | 55 | 60% |
| Woman | 37 | 40% |
| **Age group** | | |
| 18–34 years | 5 | 5% |
| 35–49 years | 45 | 48% |
| 50–64 years | 32 | 34% |
| 65 + | 11 | 12% |
| **Organizational Role** | | |
| Government | 20 | 22% |
| Non-governmental organization | 46 | 50% |
| University or research | 22 | 24% |
| Other | 4 | 4% |
| **Personal Role** | | |
| Academic/Research | 24 | 26% |
| Programmatic/Implementation | 57 | 61% |
| Other | 12 | 13% |

**Table 2. Stakeholder Survey–Frequency of responses to strength/barrier survey items taken from in-depth interview responses from public health practitioners in Tanzania.**

| Survey item | | 3-way scale | | | |
|---|---|---|---|---|---|
| | | Total respon-dents | Strongly agree or agree n (%) | Neither agree nor disagree n (%) | Strongly disagree or disagree n (%) |
| **Field grading of TF** | | | | | |
| Strength | TF grading is recommended by the World Health Organization for diagnosing trachoma in communities. | 81 | 79 (98) | 1 (1) | 1 (1) |
| | TF grading is a familiar method. | 81 | 71 (88) | 6 (7) | 4 (5) |
| | TF grading is acceptable to communities. | 81 | 69 (85) | 8 (10) | 4 (5) |
| | TF grading is harmless/does not cause side effects. | 81 | 66 (81) | 8 (10) | 7 (9) |
| | TF grading provides results on the spot. | 81 | 66 (81) | 9 (11) | 6 (7) |
| | TF grading is accurate. | 79 | 42 (53) | 18 (23) | 19 (24) |
| Barrier | TF grading is subjective and prone to human error. | 81 | 65 (80) | 5 (6) | 11 (14) |
| | TF grading is not as sensitive or specific as other methods. | 81 | 50 (62) | 15 (19) | 16 (20) |
| | Everting the eyes of children can be difficult. | 80 | 49 (61) | 15 (19) | 16 (20) |
| **Conjunctival photography** | | | | | |
| Strength | Multiple graders can grade the same photo, which will increase accuracy and help determine the status of "edge cases." | 73 | 61 (84) | 5 (7) | 7 (10) |
| | Increased costs of conjunctival photography will be less than those for eye swabs or blood spots. | 73 | 41 (56) | 26 (36) | 6 (8) |
| | The potential to use Artificial Intelligence to grade photos will provide more accurate results than field grading of TF. | 71 | 34 (48) | 30 (42) | 7 (10) |
| | The photo can be used to determine the presence or absence of other eye diseases. | 73 | 30 (41) | 25 (34) | 18 (25) |
| | Conjunctival photography is easy to implement. | 72 | 29 (40) | 21 (29) | 22 (31) |
| | Grading the photo away from the field will reduce field time compared to TF grading. | 73 | 24 (33) | 23 (32) | 26 (36) |
| | Conjunctival photography is more accurate than field grading of TF. | 73 | 21 (29) | 33 (45) | 19 (26) |
| | Conjunctival photography is more sensitive than field grading of TF. | 72 | 18 (25) | 35 (49) | 19 (26) |
| Barrier | Increased costs compared to TF grading due to need for camera equipment. | 72 | 43 (60) | 19 (26) | 10 (14) |
| | Increased costs compared to TF grading due to increased personnel training/time needed. | 72 | 41 (57) | 14 (19) | 17 (24) |
| | Community participation may be low if participants do not receive individual results. | 71 | 39 (55) | 16 (23) | 16 (23) |
| | Community participation may be low due to fear/mistrust of having photo taken. | 72 | 38 (53) | 15 (21) | 19 (26) |
| | Conjunctival photography grading is subjective and prone to human error. | 72 | 36 (50) | 21 (29) | 15 (21) |
| | Community participation may be low due to fear of side effects due to photo flash. | 72 | 27 (38) | 18 (25) | 27 (38) |
| | Conjunctival photography is perceived as less accurate compared to other methods. | 73 | 18 (25) | 38 (52) | 17 (23) |
| **Infection testing** | | | | | |
| Strength | Infection testing can detect pre-symptomatic trachoma. | 68 | 55 (81) | 9 (13) | 4 (6) |
| | Eye swabs can be tested for multiple pathogens/diseases. | 70 | 55 (79) | 11 (16) | 4 (6) |
| | Infection testing is more accurate than field grading of TF at identifying trachoma. | 70 | 54 (77) | 8 (11) | 8 (11) |
| | Infection testing is more accurate than other methods at identifying trachoma. | 70 | 52 (74) | 11 (16) | 7 (10) |
| | Eye swabs/infection testing are acceptable to communities as long as education is provided. | 69 | 51 (74) | 14 (20) | 4 (6) |
| | Eye swabs have a low chance of causing side effects. | 70 | 51 (73) | 14 (20) | 5 (7) |

*(Continued)*

**Table 2.** (Continued)

| Survey item | | 3-way scale | | | |
| --- | --- | --- | --- | --- | --- |
| | | Total respondents | Strongly agree or agree n (%) | Neither agree nor disagree n (%) | Strongly disagree or disagree n (%) |
| Barrier | Extra work needed to plan for logistics of field collection and sample processing. | 71 | 68 (96) | 1 (1) | 2 (3) |
| | Desire for samples to be tested by nearest lab may be a challenge. | 71 | 63 (89) | 4 (6) | 4 (6) |
| | Increased costs compared to TF field grading due to storage of samples. | 71 | 62 (87) | 3 (4) | 6 (8) |
| | Increased costs compared to TF field grading due to equipment needed. | 71 | 62 (87) | 5 (7) | 4 (6) |
| | Requirement for cold chain a challenge due to need for stable electricity. | 70 | 61 (87) | 4 (6) | 5 (7) |
| | Increased costs compared to TF field grading due to increased personnel training/time needed. | 70 | 48 (69) | 10 (14) | 12 (17) |
| | Poorly trained personnel might cause damage to eyes of participants. | 71 | 40 (56) | 15 (21) | 16 (23) |
| | Community participation may be low if participants do not receive individual results. | 70 | 36 (51) | 19 (27) | 15 (21) |
| | Community participation may be low due to fear of sample collection process or side effects. | 70 | 34 (49) | 15 (21) | 21 (30) |
| | Infection testing may be less accurate than TF field grading due to time between collection and testing. | 71 | 18 (25) | 12 (17) | 41 (58) |
| **Serology** | | | | | |
| Strength | Blood spots (for other diseases) are familiar to health systems. | 66 | 51 (77) | 11 (17) | 4 (6) |
| | Blood spots are easy to collect. | 66 | 47 (71) | 11 (17) | 8 (12) |
| | Blood spots are easier to transport and store than eye swabs. | 64 | 42 (66) | 13 (20) | 9 (14) |
| | Blood spots (for other diseases) are familiar and acceptable to community members. | 66 | 39 (59) | 14 (21) | 13 (20) |
| | Serological testing is more accurate than TF grading at identifying trachoma. | 66 | 29 (44) | 17 (26) | 20 (30) |
| | Serological testing is fast. | 65 | 26 (40) | 16 (25) | 23 (35) |
| | Serological testing can detect pre-symptomatic trachoma. | 66 | 21 (32) | 19 (29) | 26 (39) |
| Barrier | Extra work needed to plan for logistics of field collection and sample processing. | 66 | 58 (88) | 4 (6) | 4 (6) |
| | Increased costs compared to TF field grading due to equipment needed. | 66 | 55 (83) | 8 (12) | 3 (5) |
| | Lack of current lab capacity. | 65 | 52 (80) | 9 (14) | 4 (6) |
| | Increased costs compared to TF field grading due to increased personnel training/time needed. | 66 | 45 (68) | 10 (15) | 11 (17) |
| | Increased costs due to storage of samples. | 65 | 41 (63) | 11 (17) | 13 (20) |
| | Increased danger of working with blood/risk of medical errors, compared to other diagnostics. | 66 | 38 (58) | 15 (23) | 13 (20) |
| | Field conditions challenging, samples may be destroyed in the field. | 66 | 35 (53) | 22 (33) | 9 (14) |
| | Community participation may be low due to misunderstanding or mistrust of test purpose. | 66 | 35 (53) | 16 (24) | 15 (23) |
| | Community participation may be low if participants do not receive individual results. | 64 | 31 (48) | 13 (20) | 20 (31) |
| | Community participation may be low due to fear of finger pricking. | 66 | 28 (42) | 16 (24) | 22 (33) |

TF = trachomatous inflammation—follicular

needed equipment (62, 87%), as well as the challenge posed by the requirement for maintaining a cold chain (61, 87%). In contrast to items about feasibility, there were lower levels of agreement with barriers related to low community participation, due either to a lack of receiving individual results (36, 51%), or fear of the sample collection process or side effects (34, 49%).

The most agreed-with strengths of serology include the familiarity of health systems with blood spots (for other diseases) (51, 77%), as well as an acknowledgement that blood spots are both easy to collect (47, 71%), and easier to transport and store compared to eye swabs (42, 66%). The most agreed-with barriers to the implementation of serology included the extra work needed to plan for the logistics of field collection and sample processing (58, 88%), the increased costs compared to TF field grading due to equipment needed (55, 83%), and the lack of current lab capacity (52, 80%). Less than half of respondents agreed that community participation may be low if participants do not receive individual test results (31, 48%) or due to a fear of finger pricking (28, 42%).

The results of the comparison of responses to barriers related to community acceptability by country staff vs. other stakeholders are presented in Table 3. (In this analysis, a higher score indicates a higher level of agreement that the statement is a barrier to implementation of the indicator.) On average, the country staff identified conjunctival photography to face the greatest barriers related to community acceptability; in contrast, the other stakeholders identified serology as facing the greatest barriers related to community acceptability. However, none of the comparisons were significant at an alpha of 0.05.

Key knowledge gaps entered by respondents are presented as S5 File. Categories of knowledge gaps identified include: setting and interpreting meaningful thresholds using complementary indicators, understanding test characteristics (e.g. sensitivity and specificity) of each indicator, refining survey design to allow the inclusion of complementary indicators, interpreting serological data, understanding clinical signs and the progression of disease, integrating complementary indicators for trachoma into existing systems, understanding the acceptability and feasibility of complementary indicators, and identifying and mitigating potential issues with training and lab quality assurance.

## Discussion

From the FGDs, we found that each diagnostic method explored was generally acceptable to community members in Tanzania, with key themes such as not wanting to suffer harm from

**Table 3. Stakeholder agreement with proposed barriers of complementary indicators to trachomatous inflammation—follicular detection by affiliation (country staff vs. other stakeholders).**

| Barrier | Country staff[a] (n = 14) | Other stakeholders (n = 59)[b] | |
|---|---|---|---|
| | median (IQR) | median (IQR) | p value[c] |
| **Conjunctival photography** | | | |
| Community participation may be low due to fear/mistrust of having photo taken. | 4.0 (3.6–4.4) | 3.0 (2.0–4.0) | 0.13 |
| Community participation may be low due to fear of side effects due to photo flash. | 3.5 (2.5–4.5) | 3.0 (2.0–4.0) | 0.46 |
| Community participation may be low if participants do not receive individual results. | 4.0 (3.5–4.5) | 4.0 (3.5–4.5) | 0.37 |
| **Infection testing** | | | |
| Community participation may be low due to fear of sample collection process or side effects. | 3.0 (2.0–4.0) | 3.5 (2.5–4.5) | 0.70 |
| Community participation may be low if participants do not receive individual results. | 4.0 (3.4–4.6) | 3.5 (3.0–4.0) | 0.67 |
| **Serology** | | | |
| Community participation may be low due to fear of finger pricking. | 2.0 (1.1–2.9) | 3.0 (2.0–4.0) | 0.08 |
| Community participation may be low due to misunderstanding or mistrust of test purpose. | 3.0 (2.1–3.9) | 4.0 (3.5–4.5) | 0.38 |
| Community participation may be low if participants do not receive individual results. | 2.5 (1.6–3.4) | 4.0 (3.0–5.0) | 0.14 |

IQR = Interquartile range, a: organizational role = "Government" and personal role = "Programmatic/Implementation", b: not all respondents answered every question, c: as measured by a 2-tailed Wilcoxon-rank-sum test.

Note: A higher score indicates a stronger level of agreement that the statement is a barrier to implementation of the indicator.

the diagnostic process and the importance of (perceived) test accuracy apparent. From the IDIs with public health practitioners in Tanzania, we found that this group also considered each diagnostic method to be acceptable to community members, as long as appropriate education on the risks and benefits of participation is provided.

When global stakeholders were asked to indicate their agreement that given barriers would hinder implementation of complementary indicators, the highest agreement items tended to be those around feasibility, rather than acceptability. The highest consensus of any strength or barrier listed was agreement that the WHO recommendation for field grading of TF is a benefit of this method, with 98% of respondents in agreement. When results around community acceptability were stratified based on country staff vs. other stakeholder, no statistically significant differences were found, indicating broad agreement among respondents in the stakeholder survey that community acceptability of each method is unlikely to hinder implementation. This was concordant with the findings from community members in Tanzania, who rated each test type generally acceptable.

A study in the Bijagós Islands, Guinea Bissau on different diagnostic tests and sample types for trachoma surveillance (including clinical exams, eye swabs, and finger-prick blood samples) reported that although all the studied test types were generally acceptable, there was a preference among community members for laboratory-based testing; these results were considered more accurate than clinical examination. [26] Additionally, sample types that did not require close proximity to the eye (i.e., finger-prick blood samples) were preferred. [26] Our study had similar results, with participants noting a perception of improved accuracy from laboratory-based methods, although expressing some unease with the eye swab and blood spot procedures, especially with regards to children. This similarity in findings is reassuring given the different locations and population types, indicating some generalizability of the results. Considered in parallel, these results suggest that acceptability would not be a widespread barrier to implementation.

Different studies have shown varying levels of acceptability of blood collection. Research in sub-Saharan Africa has shown that collecting blood or other bodily samples from individuals can be challenging, with fears related to blood-stealing and intentional spreading of disease concerning public health interventions documented since colonial times. [27] Individual reticence to participation in a trial in The Gambia involving blood collection via finger stick for malaria screening was linked to fear of blood-taking due to depletion of life force and fear of exploitation. [28] In contrast, a study on the feasibility of bloodspot collection for the surveillance of human African trypanosomiasis in the Democratic Republic of the Congo found that refusal to participate in bloodspot collection was rare, although reasons for non-participation were not analysed. [29] Finger-prick blood samples are routinely taken for malaria testing, [30] which is commonplace in most places that are also trachoma-endemic. [31] Our community member participants expressed broad acceptance of blood spots, with two notable exceptions.

First, among the Maasai ethnic group one FGD (out of four) noted a general cultural dislike of blood-giving. This may be related to a noted lack of trust of non-Maasai visitors, an incomplete understanding of the purpose of interventions, and a perception of misalignment between community and government priorities, which has eroded trust in this institution. [32] A study conducted in the Sinya Ward (a Maasai community in Longido District) found that, while this community does perceive trachoma as a problem, [33] competing priorities can make it difficult to participate in trachoma-related interventions, such as MDA. [32] A lack of sensitization around MDA has previously been associated with poor uptake of azithromycin in this area, [34] underscoring the need for effective messaging on health programs in these and

other under-served communities, which, as we near trachoma elimination, may represent the last foci of TF.

Second, concerns explicitly about the purpose of blood tests were raised (among a variety of ethnic groups), with specific fears mentioned around testing for HIV without the participants' consent. These fears were relayed both by community participants themselves and recognized as a potential reason for low community acceptance by public health practitioners. In this regard, the familiarity and commonplace nature of blood testing is a disadvantage, as community members are aware that their blood samples could be tested for many other diseases, some of which (such as HIV) still carry a level of stigma. [35] However, familiarity with blood spots was also perceived as an advantage of this method, both by community members, who appreciated the lack of pain or harm caused by the sample procedure, and by public health practitioners, who noted experience with sample collection, processing, and testing by an array of health systems in Tanzania. The advantages of blood testing enable programs, with appropriate ethical approvals and participant consent, to benefit from the increased efficiency of testing one sample for many diseases. It will be up to country programs to adapt their testing strategies and how results will be used to reflect cultural preferences, especially in light of the comments made about fears of the confidentiality of HIV results. While public health practitioners in Tanzania did often recognize the familiarity of blood testing (for other diseases) by health systems as an advantage of this method, our study found a lack of acknowledgement of the potential utility of serology for the integration of trachoma monitoring with other disease programs. This group also recognized that a lack of local lab capacity would make the quick processing of blood samples a challenge, which was corroborated as a barrier to implementation by global stakeholders. The Global NTD road map 2021–2030 calls for concerted action to achieve integration across NTD programs and for increased investment in developing lab capacity in order to meet NTD targets. [19] The long-term benefit of investments in increasing and maintaining lab capacity of health systems for infection testing and serology should be underscored to stakeholders at all levels. A coordinated approach is needed to leverage the investments and achievements made to date across different disease programs, enabling the building, strengthening and maintenance of laboratory capacity and networks. [36]

Community members found conjunctival photography to be an acceptable method for trachoma diagnosis, noting a general familiarity with the concept of medical imaging and potential benefits of this method compared to field grading of TF. Another study on the acceptability of conjunctival photography among community members in Tanzania found similar sentiment around the utility of photography for grading borderline cases (contributing to greater overall accuracy), but noted an element of concern among community members regarding the exposure of children to a photo flash. [37] However, this fear pertained only to traditional digital single-lens reflex (DSLR) cameras, compared to photography via smartphone. [37] In addition, our study did not find that issues around photo use and confidentiality were a concern among community members. However, these issues were recognized as potential concerns by public health practitioners in our study, as well as both community members and Tropical Data trainers in the previous Tanzanian qualitative study. [37] Consent materials for activities incorporating conjunctival photography for research or programmatic use should be very clear as to the risks and benefits of participating.

We found that generally, FGDs with men were more favorable to each of the test types compared to FGDs with women, with FGDs with men more likely to have at least one participant indicate that they would hypothetically allow the children under their care to participate in each test type compared to FGDs with women. A previous systematic review and meta-analysis on the acceptance of the COVID-19 vaccine, which included studies from each WHO region, found that male gender was associated with an increased likelihood of vaccine acceptance. [38]

A systematic review on this topic in sub-Saharan Africa found similar results. [39] In terms of care-giving behavior, a multi-country questionnaire similarly found that fathers are more willing than mothers to vaccinate their children against COVID-19, possibly due to differences in risk-taking behaviors among fathers vs. mothers. [40] It may be advisable to target messaging around the safety of and risk of discomfort or adverse consequences from each diagnostic method specifically to women caregivers, especially as they are often the decision-makers in regards to children's healthcare. [41]

The high level of stakeholder agreement that a WHO recommendation of field grading of TF is beneficial indicates that additional WHO guidance around complementary indicators may be advantageous. In informal workshops, tests for infection and serology have already been advocated for by WHO, [15] and the forthcoming guidelines on their use are eagerly awaited.

Seventy unique questions regarding knowledge gaps preventing the interpretation and implementation of complementary indicators were identified by global stakeholders. The collation and categorization of these questions will hopefully serve as a useful resource for helping to define future research priorities as we approach the trachoma elimination endgame.

Our study had several limitations. Messaging as to the testing procedure and the fact that not all tests would yield individual results was included in the FGD topic guide and stated by the research assistant while conducting the FGD. However, it is clear from the transcripts that participants did not always adequately understand specifics of each test type, or that they would not always receive individual results for each sample type. This may make conclusions about the strength of acceptance difficult to directly apply to surveillance settings, where participants could not expect individual results and/or treatment. However, capturing common misconceptions by community members about each test type is also important, as it provides useful information for programs moving forward and highlights the need to focus on community sensitization, engagement, and setting participant expectations both on the test procedures and receipt of results. Furthermore, we note that asking community members if they would hypothetically participate in each sample collection type may yield different results than actually presenting participants with the opportunity to be tested. In addition, community FGDs and IDIs only took place in a selection of sites within one country. There are likely to be differing views across the trachoma-endemic world.

In attempting to evaluate the opinion of a variety of trachoma stakeholders, the stakeholder survey was disseminated to as wide an audience as possible, resulting in an opportunistic sample. While an opportunistic sample has implications for generalizability, the multiple avenues of survey dissemination, the offering of multiple languages of the survey, and the diverse geographic representation of respondents indicate that the results are likely fairly representative of trachoma stakeholders as a whole.

Strengths and barriers from the IDIs were used to formulate the questions for the stakeholder survey. In order to preserve the intent of these responses, the items in the stakeholder survey could have been perceived as overly positive (for the strengths) and negative (for the barriers), which may have had an effect in biasing stakeholders' level of agreement with each item. In addition, the factual accuracy of the statements offered by the IDIs (for example, that one test may be more "accurate" than another) is debatable, which may have hindered interpretation of participants' agreement with these items. There may have also been potential strengths and barriers to each method not identified by IDI respondents (such as the potential benefit of being able to test blood spots for multiple infections). In each case, to avoid biasing the selection of strengths and barriers presented in the questionnaire, care was taken to preserve the original content and meaning from the IDIs. However, future surveys may wish to provide more comprehensive lists of strengths and barriers and more neutral question

wording (for example, by allowing respondents to rate the acceptability of different methods, rather than their agreement with subjective statements).

Future work would be needed in order to study the opinions of communities in other regions to have more confidence in the generalizability of our results. A deeper exploration into the attitudes and perceptions of blood testing among the Maasai ethnic group, along with the effectiveness of potential education or sensitization strategies, is warranted in order to ensure this group is equitably represented in future trachoma programming. In addition, a study of non-response rates in studies where additional indicators are collected would yield information about the acceptability of different sample collection methods for trachoma detection during real-world program activities.

## Conclusions

We found that conjunctival photography, infection testing, and serology were all generally acceptable to community members in Tanzania. Critical themes included participants not wanting to suffer harm from the diagnostic process and the importance of (perceived) test accuracy. Many of the perceived disadvantages mentioned by community members in Tanzania, such as fear of undue discomfort or adverse consequences of participating in the diagnostic process or mistrust of the test purpose (for serology) are potential focus areas for outreach by healthcare workers and other partners prior to implementation. This need for community education on the risks and benefits of participation of each of the methods was underscored in interviews with public health practitioners in Tanzania. According to both Tanzanian public health practitioners and global stakeholders, questions of feasibility remain, especially for infection testing and serology. The fact that in general, all methods were acceptable to community members is positive, as it may be that each indicator has a role to play for trachoma elimination surveillance purposes.

## Supporting information

**S1 File. Topic Guides.**
(DOCX)

**S2 File. *A prioi* Codes.**
(PDF)

**S3 File. SHS Questionnaire.**
(XLSX)

**S4 File. Framework Matrix.**
(XLSX)

**S5 File. Knowledge Gaps.**
(PDF)

## Acknowledgments

The authors which to thank the participants of the focus group discussions, in-depth interviews, and the survey respondents for their involvement in this project. The authors alone are responsible for the views expressed in this article and they do not necessarily represent the views, decisions or policies of the institutions with which they are affiliated.

## Author Contributions

**Conceptualization:** Kristen K. Renneker, Anthony W. Solomon, Emma M. Harding-Esch.

**Data curation:** Kristen K. Renneker.

**Formal analysis:** Kristen K. Renneker.

**Funding acquisition:** Kristen K. Renneker.

**Investigation:** Kristen K. Renneker, Tara B. Mtuy, Patrick Mosha, Jeremiah Mepukori Mollel.

**Methodology:** Kristen K. Renneker, Tara B. Mtuy, George Kabona, PJ Hooper, Paul M. Emerson, T. Deirdre Hollingsworth, Robert Butcher, Emma M. Harding-Esch.

**Project administration:** Kristen K. Renneker, George Kabona, Stephen Gabriel Mbwambo.

**Resources:** Kristen K. Renneker.

**Software:** Kristen K. Renneker.

**Supervision:** Kristen K. Renneker, Emma M. Harding-Esch.

**Validation:** Kristen K. Renneker, Tara B. Mtuy.

**Visualization:** Kristen K. Renneker.

**Writing – original draft:** Kristen K. Renneker.

**Writing – review & editing:** Kristen K. Renneker, Tara B. Mtuy, George Kabona, Stephen Gabriel Mbwambo, Patrick Mosha, Jeremiah Mepukori Mollel, PJ Hooper, Paul M. Emerson, T. Deirdre Hollingsworth, Robert Butcher, Anthony W. Solomon, Emma M. Harding-Esch.

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
