## [Decision Letter · Decision Letter 0]

13 Feb 2024

Dear Ms. Renneker,

Thank you very much for submitting your manuscript "Acceptability and feasibility of tests for infection, serological testing and photography to define need for interventions against trachoma" for consideration at PLOS Neglected Tropical Diseases. As with all papers reviewed by the journal, your manuscript was reviewed by members of the editorial board and by several independent reviewers. The reviewers appreciated the attention to an important topic. Based on the reviews, we are likely to accept this manuscript for publication, providing that you modify the manuscript according to the review recommendations. 

Sincerely,

Stuart D. Blacksell

Section Editor

Reviewer's Responses to Questions

**Key Review Criteria Required for Acceptance?**

**Methods**

-Are the objectives of the study clearly articulated with a clear testable hypothesis stated?

-Is the study design appropriate to address the stated objectives?

-Is the population clearly described and appropriate for the hypothesis being tested?

-Is the sample size sufficient to ensure adequate power to address the hypothesis being tested?

-Were correct statistical analysis used to support conclusions?

-Are there concerns about ethical or regulatory requirements being met?

Reviewer #1: Methods are clearly articulated. The investigators progressed through a logical progression of focus groups with community members, in-depth interviews with local stakeholders in Tanzania and finally a survey of global stakeholders.

Reviewer #2: The objectives of the study are clearly defined within a clear testable approach that was given. The study design is appropriate and it addresses the main objectives. The population is clearly described and appropriate for the hypothesis being tested. The sample size is sufficient to ensure adequate power to address the hypothesis being tested.

The correct statistical method was used to support the conclusions. I have no concerns about the ethical or regulatory requirements obtained by the authors of the study.

Reviewer #3: Yes

**Results**

-Does the analysis presented match the analysis plan?

-Are the results clearly and completely presented?

-Are the figures (Tables, Images) of sufficient quality for clarity?

Reviewer #1: Results and data analyses are clear. The reader will likely need to review the methods and results concurrently/iteratively since the results from the earlier phases influence the methods of the subsequent phases, but this is unavoidable with this type of study.

Reviewer #2: The analysis presented matches the analysis plan and the results are clear and complete. The tables and figures are of high quality and sufficient.

Reviewer #3: Yes

**Conclusions**

-Are the conclusions supported by the data presented?

-Are the limitations of analysis clearly described?

-Do the authors discuss how these data can be helpful to advance our understanding of the topic under study?

-Is public health relevance addressed?

Reviewer #1: Conclusions are supported by the data. The authors recognize the major limitation of this design ("ask a hypothetical question, get a hypothetical answer"), but deeply engage with the results to explore ways to improve acceptability of any future interventions.

Reviewer #2: The conclusions are supported by the data presented and the limitations of the analysis are clearly described. The authors have discussed how this data can benefit and help to understand trachoma testing worldwide. The public health relevance is addressed and I can easily see work from this study being implemented in many Countries of similar settings.

Reviewer #3: Yes

**Editorial and Data Presentation Modifications?**

Reviewer #1: Minor change, in the first paragraph of discussion, the authors refer to the IDI respondents as "health workers," but I would recommend they continue to use public health practitioners, as "health worker" sounds similar to "healthcare worker" and this was only 1 of 11 of this group.

Reviewer #2: "Accept"

Reviewer #3: (No Response)

**Summary and General Comments**

Reviewer #1: This is an important contribution to the trachoma field that reports results of qualitative and quantitative

surveys seeking to explore the acceptability of new survey tools. The authors find that three new methods (photography, eye swab for active infection, finger prick for serology) seem to be reasonably acceptable without major barriers to adoption. The authors do note several factors that may improve acceptability (ie education about long term risks, reassurance/trust about the lack of HIV testing, etc).

My main comments center around the utility of answers from the focus groups which seem to be based on inaccurate on contradictory assumptions. For example, the respondents voiced support of photography since it could be quicker than in person exam, but I am not sure that this is true, as the lid must still be everted and then the correct exposure must be acquired. Likewise, a major concern about photography seemed to be the flash, but not all programs have used a flash. The authors indicate that the timing of results (ie delayed and not directly communicated to individuals) was not always conveyed to focus group participants, but whether the authors believe this would have a major effect on their inferences is not completely explored. Finally, assumptions about the accuracy and modernity/legitimacy about infection/serology testing do not seem completely accurate as discussed by focus group participants. The authors indicate that these tests are complementary, but there seems to be an implied preference for one test over another, rather than people being exposed to a possible battery of tests. 

Overall, with all of this, I think a little more treatment in the discussion about the role of (gauging/setting) expectations of a given new modality would be helpful, as it seems clear that positive sentiments (ie about the legitimacy of medical tests) might benefit an initial survey, but if other expectations are not met (about the immediacy of test results), this could severely hamper subsequent surveys. 

Also, the advantage of monitoring for other diseases is an advantage of the medical tests, which the authors imply was under-appreciated by respondents. It is unclear how to balance this advantage with legitimate concerns about the samples being used to detect unrelated diseases (e.g., HIV). How would the authors recommend balancing this in a program?

Reviewer #2: The study design is appropriate and addressed perfectly the feasibility and accessibility of the use of alternative indicators for trachoma. It is novel, what was unique in the study to me was the comparison between the Country participants and international stakeholders. I do not have any concerns about dual publication, research or publication ethics.

Reviewer #3: A mixed methods study about the acceptability of conjunctival photography, conjunctival swabbing, and serology for trachoma surveillance. Overall I found this to be well designed and well reported. I just have a couple very minor comments:

Table 2 was cut off on the right hand side

Line 433: minor comment but the mention of “agreement” made me think this analysis was assessing some measure of inter-respondent agreement. But here I think this is simply a stratified analysis? I might word it a little differently, saying that on average the country staff identified conjunctival photography to face the greatest barriers to community acceptability, and that in contrast, the other stakeholders identified serology as facing the greatest barriers

A minor comment: there have been many trachoma studies conducted throughout the world that have incorporated conjunctival photography, conjunctival swabs, and serology, some of which have sampled the same participants longitudinally over years. I am not aware of massive refusals of any of these testing modalities in these studies, which also lends some support for the idea that all three of these methods are acceptable to community members.

PLOS authors have the option to publish the peer review history of their article (what does this mean?). If published, this will include your full peer review and any attached files.

Reviewer #1: No

Reviewer #2: Yes: Titus Watitu Kimani

Reviewer #3: No

Figure Files:

Data Requirements:

Please note that, as a condition of publication, PLOS' data policy requires that you make available all data used to draw the conclusions outlined in your manuscript. Data must be deposited in an appropriate repository, included within the body of the manuscript, or uploaded as supporting information. This includes all numerical values that were used to generate graphs, histograms etc.. For an example see here: http://www.plosbiology.org/article/info:doi%2F10.1371%2Fjournal.pbio.1001908#s5.

Reproducibility:

References

---

## [Decision Letter · Decision Letter 1]

13 May 2024

Dear Ms. Renneker,

We are pleased to inform you that your manuscript 'Acceptability and feasibility of tests for infection, serological testing and photography to define need for interventions against trachoma' has been provisionally accepted for publication in PLOS Neglected Tropical Diseases.

Best regards,

Stuart D. Blacksell

Section Editor

Reviewer's Responses to Questions

**Key Review Criteria Required for Acceptance?**

**Methods**

-Are the objectives of the study clearly articulated with a clear testable hypothesis stated?

-Is the study design appropriate to address the stated objectives?

-Is the population clearly described and appropriate for the hypothesis being tested?

-Is the sample size sufficient to ensure adequate power to address the hypothesis being tested?

-Were correct statistical analysis used to support conclusions?

-Are there concerns about ethical or regulatory requirements being met?

Reviewer #1: all my prior comments have been satisfactorily addressed.

Reviewer #2: The methods section is detailed and well-structured, providing clear information on the study design, ethical considerations, participant selection, and data collection methods.

It would be helpful if possible to include more information on the specific questions asked during the focus group discussions and in-depth interviews, particularly regarding participants' perceptions of different diagnostic approaches.

Reviewer #3: yes

**Results**

-Does the analysis presented match the analysis plan?

-Are the results clearly and completely presented?

-Are the figures (Tables, Images) of sufficient quality for clarity?

Reviewer #1: all my prior comments have been satisfactorily addressed.

Reviewer #2: The results section effectively presents the findings of both the qualitative and quantitative components of the study.

Consider providing more specific details or examples of the themes identified in the focus group discussions and in-depth interviews to illustrate participants' perspectives on the acceptability and feasibility of complementary indicators.

Reviewer #3: yes

**Conclusions**

-Are the conclusions supported by the data presented?

-Are the limitations of analysis clearly described?

-Do the authors discuss how these data can be helpful to advance our understanding of the topic under study?

-Is public health relevance addressed?

Reviewer #1: all my prior comments have been satisfactorily addressed.

Reviewer #2: Conclusions are supported by the data presented

Limitations are clearly described and yes the public health relevancy is addressed.

Reviewer #3: yes

**Editorial and Data Presentation Modifications?**

Reviewer #1: all my prior comments have been satisfactorily addressed.

Reviewer #2: in the section - Keywords: trachoma; conjunctival photography; Serology; infection testing; acceptability;

feasibility # Consider use small letter for serology

Line 232- no response rate could be calculated -# is it could or could not in this case, because the survey was distributed online and could potentially be accessed by anyone worldwide with internet access, the researchers didn't have a clear idea of the total number of people who could have potentially participated. Without knowing this total target population, they couldn't calculate a response rate.

Reviewer #3: (No Response)

**Summary and General Comments**

Reviewer #1: all my prior comments have been satisfactorily addressed.

Reviewer #2: Overall, the manuscript is well-structured and very informative.

Reviewer #3: The authors addressed my comments/suggestions.

PLOS authors have the option to publish the peer review history of their article (what does this mean?). If published, this will include your full peer review and any attached files.

Reviewer #1: No

Reviewer #2: **Yes: **TITUS WATITU KIMANI

Reviewer #3: No

---

## [Editor Report · Acceptance letter]

30 May 2024

Dear Ms. Renneker,

We are delighted to inform you that your manuscript, "Acceptability and feasibility of tests for infection, serological testing, and photography to define need for interventions against trachoma," has been formally accepted for publication in PLOS Neglected Tropical Diseases.

Best regards,

Shaden Kamhawi

co-Editor-in-Chief

Paul Brindley

co-Editor-in-Chief
